# Multicolor hyperafterglow from isolated fluorescence chromophores

Xiao Zhang[1], Mingjian Zeng[1], Yewen Zhang[1], Chenyu Zhang[1], Zhisheng Gao[1], Fei He[1], Xudong Xue[1], Huanhuan Li[1], Ping Li[1], Gaozhan Xie[1], Hui Li[1], Xin Zhang[1], Ningning Guo[1], He Cheng[1], Ansheng Luo[1], Wei Zhao[1], Yizhou Zhang[2], Ye Tao [1]✉, Runfeng Chen [1]✉ & Wei Huang [1,3]✉

High-efficiency narrowband emission is always in the central role of organic optoelectronic display applications. However, the development of organic afterglow materials with sufficient color purity and high quantum efficiency for hyperafterglow is still great challenging due to the large structural relaxation and severe non-radiative decay of triplet excitons. Here we demonstrate a simple yet efficient strategy to achieve hyperafterglow emission through sensitizing and stabilizing isolated fluorescence chromophores by integrating multi-resonance fluorescence chromophores into afterglow host in a single-component copolymer. Bright multicolor hyperafterglow with maximum photoluminescent efficiencies of 88.9%, minimum full-width at half-maximums (FWHMs) of 38 nm and ultralong lifetimes of 1.64 s under ambient conditions are achieved. With this facilely designed polymer, a large-area hyperafterglow display panel was fabricated. By virtue of narrow emission band and high luminescent efficiency, the hyperafterglow presents a significant technological advance in developing highly efficient organic afterglow materials and extends the domain to new applications.

Purely organic afterglow materials have attracted an exponential amount of attention in photonics and electronics fields, which have been witnessed by remarkable successes in broad application areas spanning from information anticounterfeiting, and sensors to afterglow displays and bio-/X-ray imaging[1–16]. The impressive charm of organic afterglow materials relies on the fact that the significant breakthrough of the modulation of triplet state natures for on-demand achieving outstanding properties and functionalities through easy chemical modification with endless possibilities, revolutionizing the immanent understandings of the excited state characters originated from purely organic functional materials[17–22]. Despite fascinating prospects in achieving color-tunable afterglow emission and ultralong lifetimes[23–30], the current development of organic afterglow emission has been largely impeded by the broad emission peak associated with

the inherent large structural relaxation at the triplet ($T_1$) state and by the low photoluminescence quantum yield (PLQY) that attributed to the weak radiative decay of spin-forbidden $T_1$ excitons; and the limited color purity and low PLQY, in turn, place the constraints on organic afterglow materials in the application of high-resolution and wide color gamut afterglow displays[31,32]. Therefore, such intrinsic flaws spontaneously stimulate the exploration of applicable construction strategies to develop organic afterglow systems with simultaneously sharp photoluminescence spectrum and high PLQY.

Multi-resonance (MR) chromophore is a new type of fused polycyclic aromatic framework featuring regular arrangements of the electron-donating atom and electron-withdrawing atoms, which empower effectively complementary MR effect[33–37]. This MR effect enables separated frontier molecular orbitals (FMOs) that are localized

[1]State Key Laboratory of Organic Electronics and Information Displays & Institute of Advanced Materials (IAM), Nanjing University of Posts & Telecommunications, 210023 Nanjing, China. [2]Institute of Advanced Materials and Flexible Electronics (IAMFE) Nanjing University of Information Science and Technology, 210044 Nanjing, China. [3]Frontiers Science Center for Flexible Electronics (FSCFE), Institute of Flexible Electronics (IFE), Northwestern Polytechnical University, 710072 Xi'an, China. ✉e-mail: iamytao@njupt.edu.cn; iamrfchen@njupt.edu.cn; iamwhuang@njtech.edu.cn

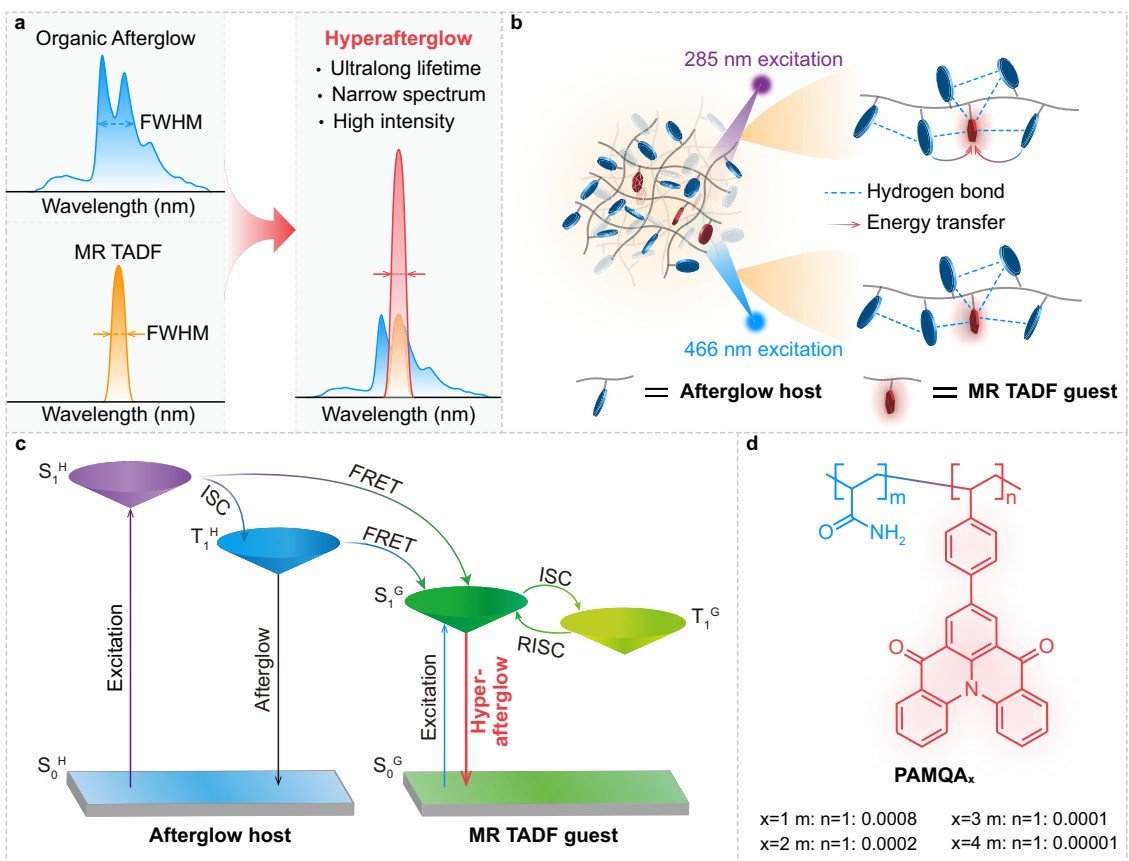

**Fig. 1 | Schematic illustration of the hyperafterglow and rational design strategy of hyperafterglow polymers. a** Spectra variation of conventional organic afterglow and hyperafterglow emission. **b** Molecular design of amorphous hyperafterglow polymers through the integration of MR TADF guest (emitters) into rigid afterglow polymeric host (sensitizers). **c** Proposed luminescent mechanism for achieving hyperafterglow emission through afterglow host-sensitized MR TADF emission. (FRET: Förster resonance energy transfer). Noted that the non-radiative decay was omitted. **d** Molecular structures of developed hyperafterglow polymers (PAMQA$_x$).

on different atoms to minimize the singlet–triplet energy gap for the facilitation of thermally activated delayed fluorescence (TADF) through reversed intersystem crossing (RISC) and to weaken bonding/antibonding attribute between different atoms for the suppression of vibrational relaxation, thus conferring theoretically 100% exciton utilization and narrow emission band[38]. However, the highly planar molecular framework of MR molecules is apt to form effective molecular packing, thus resulting in disreputable aggregation-caused quenching (ACQ) phenomena and emission bandwidth broadening. Notably, a one-stone-two-birds TADF sensitized strategy, namely hyperfluorescence[39–43], is proposed by Duan and Adachi to realize narrowband emission with 100% exciton utilization through doping a small amount of MR emitters into TADF sensitizer; for this sensitized process, TADF emitters not only serve as energy donor to utilize 100% excitons but also act as the matrix to confine the MR emitters in single molecule state for maintaining narrow bandwidth emission and high luminescence intensity. Vitalized by this hyperfluorescence, we speculate that, a new afterglow emission mechanism, named hyperafterglow, can be proposed to develop organic afterglow materials from isolated chromophores showing narrow bandwidth emission spectrum and high PLQY (Fig. 1a).

## Results

### Material design and synthesis

To verify our envision, we present a universal strategy to achieve high-efficiency hyperafterglow through stabilizing and sensitizing the isolated MR TADF chromophores in copolymer systems under ambient conditions (Fig. 1b, c). In this copolymer, polyacrylamide (PAM) was

selected as the polymeric matrix and afterglow sensitizer, because its rich carbonyl groups and amino groups could not only form a rigid polymer hydrogen bond network to reduce non-radiative decay (Fig. 1b) but also promote singlet-to-triplet intersystem crossing (ISC) to achieve excellent afterglow emission[44–46]; meanwhile, a MR chromophore, named 7-(4-vinylphenyl)quinolino3,2,1-deacridine-5,9-dione (VQA) with amine/carbonyl multi-resonance[36,47], is covalently integrated into the PAM chains to serve as narrowband emitters for enabling hyperafterglow through effective energy transfer from PAM to VQA (Fig. 1c). As a proof of concept, we exploited a set of copolymers (PAMQA$_x$, X = 1, 2, 3 and 4) by radical copolymerization of acrylamide (AM) and VQA with the molar fed ratios of 1:0.0008 (PAMQA$_1$), 1:0.0002 (PAMQA$_2$),1:0.0001 (PAMQA$_3$) and 1:0.00001 (PAMQA$_4$), respectively (Fig. 1d). The amorphous polymeric structures were systematically established by nuclear magnetic resonance (NMR), Fourier transform infrared spectroscopy (FTIR) as well as gel permeation chromatography (GPC) and powder XRD measurements (Supplementary Figs. 1–10). Compared to the physically mixed PAM and VQA (Supplementary Fig. 7) showing obvious chemical shifts from the VQA monomer, the $^1$H NMR spectra of PAMQA$_x$ confirm that VQA should have indeed participated in the copolymers.

### Hyperafterglow properties

Expectantly, bright long-lived pure-green luminescence from PAMQA$_x$ films exhibiting FWHMs of ~38 nm, high PLQYs up to 88.9%, and a lifetime of up to 65.9 ms can be easily observed after the removal of UV light (Fig. 2a–c, Supplementary Figs. 11–13 and Supplementary Table 2). These results experimentally confirm the achievement of

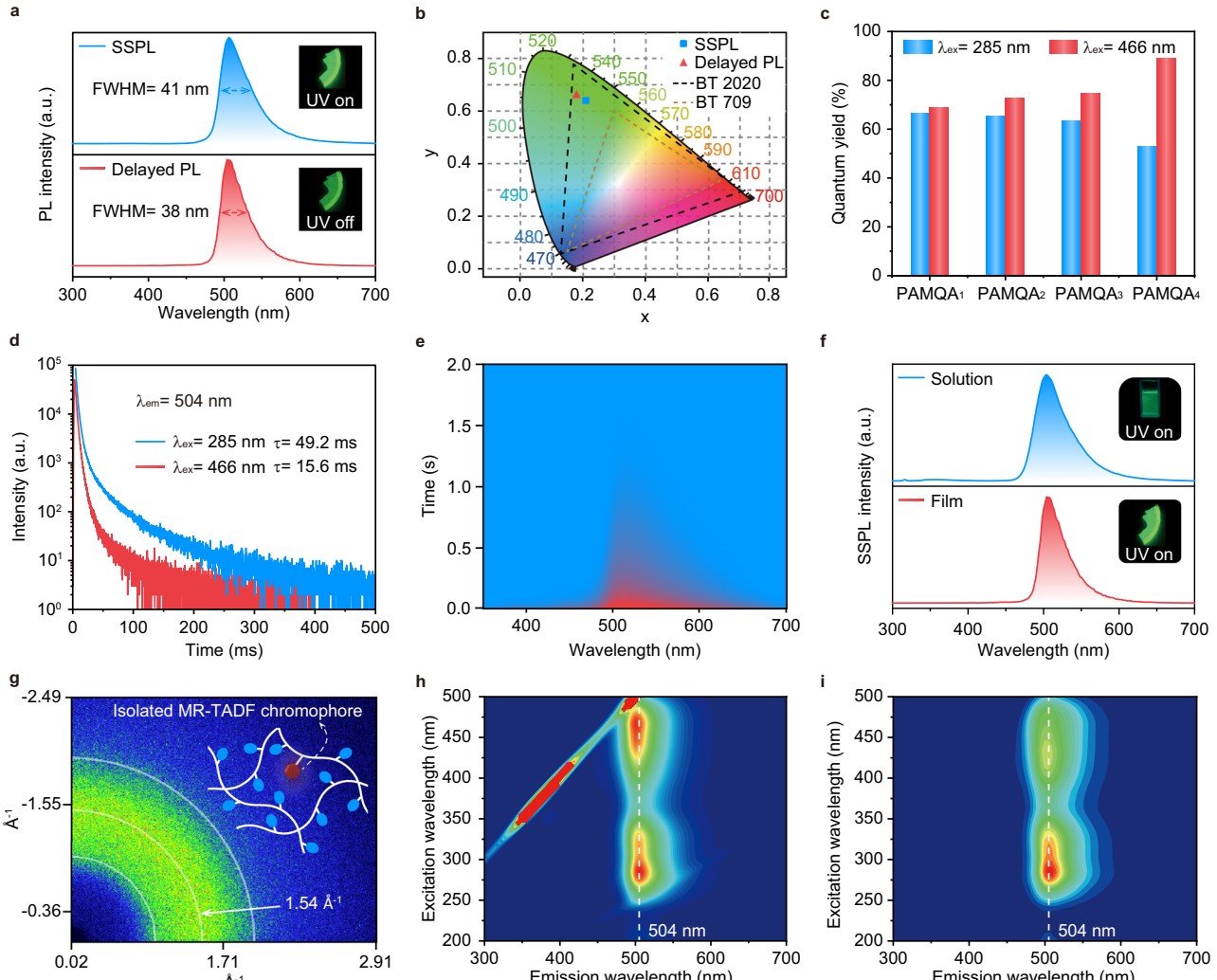

**Fig. 2 | Photophysical properties of hyperafterglow polymers under ambient conditions. a** Normalized steady-state PL (SSPL, blue line) and delayed PL (10 ms delay) spectra (red line) of PAMQA$_3$ film ($\lambda_{ex}$ = 285 nm). Inset: photographs of PAMQA$_3$ film taken upon turning on (top panel) and off (bottom panel) a 285 nm UV lamp. **b** CIE chromaticity diagram for SSPL and delayed PL emission of PAMQA$_3$ ($\lambda_{ex}$ = 285 nm). **c** PLQYs of PAMQA$_x$ films upon 285 and 466 nm excitation. **d** Lifetime decay profiles of emission band at 504 nm of PAMQA$_3$ film upon 285 and 466 nm excitation. **e** Transient emission decay images of PAMQA$_3$ film ($\lambda_{ex}$ = 285 nm). **f** Normalized SSPL spectra of PAMQA$_3$ aqueous solution (top panel) and film (bottom panel). **g** 2D-WAXS pattern of PAMQA$_3$ film. The inset shows a schematic illustration of the isolated afterglow from a single-component copolymer. **h, i.** Excitation-SSPL (**h**) and excitation-delayed PL (**i**) (25 ms delay) mappings of PAMQA$_3$ film.

hyperafterglow emission for the first time (Supplementary Fig. 14). And, with the decreasing feeding ratio of VQA moiety, the main emission peaks of PAMQA$_x$ films are slightly hypochromatic shift, potentially indicating the decreased intermolecular interaction between VQA moiety for boosting the single molecule emission characteristic (Supplementary Fig. 11). It should be noted that, for PAMQA$_4$ film with a feed ratio of 1:0.00001, besides the main emission peak at 496 nm, the additional emission profiles from PAM were also observed. The results indicate that the feed ratio between AM and VQA should be carefully modulated.

To acquire a deeper understanding of this hyperafterglow emission, we then performed a series of photophysical measurements using PAMQA$_3$ as a model polymer. As shown in Fig. 2a, the identical steady-state PL (SSPL) and delayed PL spectra of PAMQA$_3$ film at around 504 nm with FWHM of ~38 nm were recorded, which demonstrates the similar exciton decay process under and cease 285 nm UV excitation. Compared to the delayed PL spectra, the mixed emission species from prompted fluorescence and delayed fluorescence leads to the slightly larger FWHM in SSPL spectra[13]. The Commission

International de l'Eclairage (CIE) coordinates were calculated to be (0.21,0.64) and (0.18,0.66) for the SSPL and delayed PL emission of PAMQA$_3$ film, which exceed the color gamut of BT 709 and are quite close to the frontier of BT 2020[48], holding great promise to broaden the emission gamut of afterglow displays (Fig. 2b). Time-resolved emission spectra demonstrated a long-lived and stable hyperafterglow emission showing ~3 nm FWHMs variation within the increasing delayed time (Fig. 2d, e and Supplementary Fig. 15).

To grape the source and demonstrate the TADF feature of hyperafterglow emission in the developed polymeric system, the photophysical attributes of VQA monomer were studied in detail. The small value of $\Delta E_{ST}$ of 0.2 eV and sharp emission at 468 nm with FWHM of 25 nm (0.14 eV) in toluene demonstrated that VQA should be an MR-TADF chromophore (Supplementary Fig. 16)[49], which was experimentally confirmed by the short-range charge transfer (ICT) transition showing charge transfer absorption bands at around ~460 nm and red-shifted SSPL spectra (Supplementary Fig. 17) as well as theoretically certificated by separated FMOs distributed on different atom (Supplementary Fig. 18). The concentrations dependent emission nature

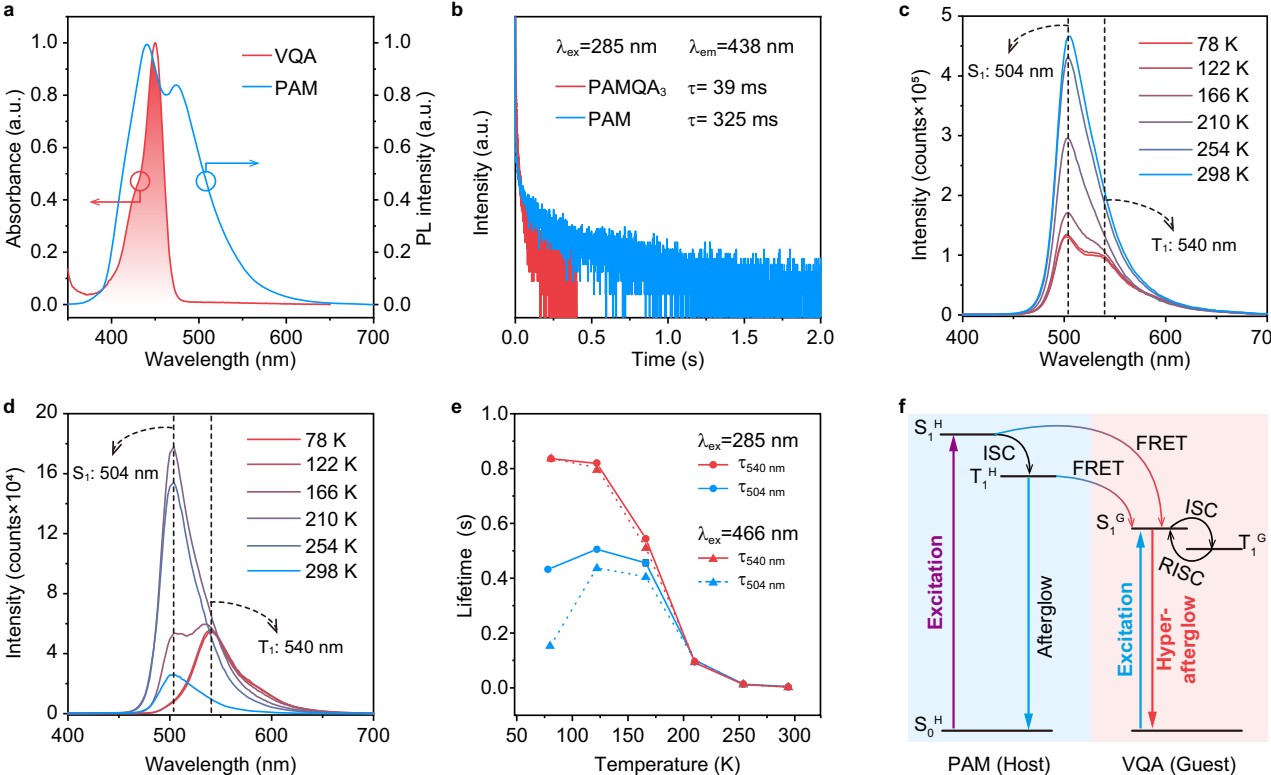

**Fig. 3 | Mechanism investigations of hyperafterglow emission. a** Delayed PL spectrum (10 ms delay) of PAM film and absorption spectrum of VQA solution. **b** Lifetime decay profile of emission band at ~438 nm of PAM and PAMQA$_3$ films under 285 nm UV light excitation. **c, d** Temperature-dependent SSPL (**c**) and delayed PL (10 ms delay) (**d**) spectra from 78 to 298 K of PAMQA$_3$ film. **e** Temperature-dependent lifetime variations of emission bands at ~504 and 540 nm of PAMQA$_3$ film under 285 and 466 nm excitation from 78 to 298 K. **f** Proposed luminescent mechanism of the hyperafterglow excited by 285 and 466 nm. Noted that the fluorescence and non-radiative process were omitted to clearly demonstrate the hyperafterglow.

suggests its high aggregation tendency owing to its rigid and planar molecular configuration (Supplementary Fig. 19). The delay components of 33.7 μs can be observed in transient fluorescence decay curves of VQA solution, which further confirms the TADF character of VQA (Supplementary Fig. 20). Through doping VQA into PAM matrix (1 wt‰), a largely red-shifted emission band at ~500 nm was found (Supplementary Fig. 18b), indicating that the hyperafterglow emission of PAMQA$_3$ should derive from the VQA. Notably, the copolymerization is much more effective than the physically blended polymer system of VQA and PAM to achieve an ultralong lifetime of hyperafterglow emission (Supplementary Fig. 21).

To verify the distinct hyperafterglow that originated from the isolated MR-TADF chromophore of VQA, we performed the photophysical measurements in an aqueous solution. The SSPL and delayed PL spectra of PAMQA$_3$ showed an obvious pure-green emission at 504 nm in an aqueous solution, which is in accordance with the emission behavior in film state under ambient conditions (Fig. 2f) and 77 K (Supplementary Figs. 22–24). Also, the absorption spectra in the solution state are quite similar to those in film states (Supplementary Fig. 25). These results certificated that the intermolecular interaction of the VQA unit was effectively suppressed and the hyperafterglow emission of PAMQA$_3$ film should be derived from its single-molecule state[50]. These discoveries were further certificated by wide-angle X-ray scattering patterns, showing only broad scattering bands at around 1.54 Å attributed to the scattering of PAM (Fig. 2g, Supplementary Figs. 26, 27)[51]. Although there were no intermolecular interactions between VQA moiety, the chromophore is anchored by plenty of hydrogen bonds with PAM, conferring a stiffness environment to stabilize and isolate the emissive chromophore for afterglow emission.

The excitation-delayed PL mapping of PAMQA$_3$ film is also quite similar to the excitation-SSPL mapping (Fig. 2h, i), showing two main

excitation bands located at the 240–370 nm range with a maximum excitation peak at 285 nm and at the 390–500 nm with a maximum excitation peak at 466 nm, which is in good agreement with the absorption profiles of PAMQA$_3$ film and solution (Supplementary Fig. 25) as well as VQA in solution (Supplementary Fig. 17), suggesting again the isolated emission nature of PAMQA$_3$. As revealed by Supplementary Fig. 28, the excitation-SSPL and excitation-delayed PL mappings of PAM film can be excited by 200–380 and 250–380 nm, respectively. Therefore, when the PAMQA$_3$ film was excited by 285 nm UV light, both PAM and VQA units were excited, displaying a hyperafterglow emission peaked at 504 nm; in contrast, when the PAMQA$_3$ film was excited by 466 nm visible light, only VQA unit was excited, and PAM may only act as a rigid matrix to suppress non-radiation decay for triggering hyperafterglow emission peaked at 504 nm. The afterglow lifetimes of PAMQA$_3$ film at 504 nm were 49.2 and 15.6 ms when excited by 285 and 466 nm (Fig. 2d), respectively. The emission behaviors of PAMQA$_1$, PAMQA$_2$, and PAMQA$_4$ showed a similar tendency (Supplementary Figs. 11, 12, 22, 23, and 29). Considering the combined results of the same emission spectra, different lifetimes, and PLQYs when excited by 285 nm UV light and 466 nm visible light, the different photophysical processes should be occurred, showcasing the possible sensitization process through energy transfer from PAM to VQA unit when excited by 285 nm.

## Mechanism investigations

To confirm the presence of energy transfer in PAMQA$_x$ when excited by 285 nm, we performed a set of photophysical measurements of PAM and PAMQA$_3$. As shown in Fig. 3a, the absorption of the VQA solution largely overlapped with SSPL and delayed PL spectra of PAM (Supplementary Fig. 30), maintaining a premise for facilitating singlet-singlet and triplet-singlet FRET[27,52,53]. And, for the excitation of PAMQA$_3$

film by 285 nm UV light, the FRET was experimentally verified by the time-resolved emission profiles of PAM and PAMQA$_3$, demonstrating a largely decreased lifetime of PAM when the VQA was covalently integrated into the PAM chains (Fig. 3b). Also, the SSPL and delayed PL spectra of PAMQA$_4$ with an extremely low concentration of VQA exhibited obviously delayed PL emission derived from PAM (Supplementary Figs. 11 and 22), supporting again the speculation of energy transfer process within copolymer system when excited by 285 nm UV light. PAMQA$_x$ exhibit high-energy transfer efficiencies of up to 94.7% (Supplementary Table 3), as calculated from the amplitude averaged lifetimes (438 nm) of PAM and PAMQA$_x$ (Supplementary Fig. 31). Notably, due to the inevitable exciton loss in the energy transfer process, the PLQYs excited by 285 nm were slightly lower than those excited by 466 nm (Fig. 2c).

The MR-TADF characteristic of PAMQA$_3$ was further verified by temperature-dependent SSPL and delayed PL investigations under the excitation by 285 and 466 nm. For SSPL, as the temperature decreases from 298 to 78 K, the emission bands peaked at 504 nm ascribed to S$_1$ of VQA monotonically decreased due to the suppressed RISC process from T$_1$ to S$_1$, which is a typical attribute of TADF luminogens[54]; while, the 540 nm emission band belonged to T$_1$ gradually became apparent (Fig. 3c and Supplementary Fig. 32). For the hyperafterglow emission, it can be seen that with the decrease of temperature, the emission intensities and lifetimes of 504 nm firstly increased, reached the maximum at 210 K, and then decreased from 210 to 78 K, and completely disappeared at 78 K in the delayed PL spectra (Fig. 3d, e and Supplementary Figs. 32–34); the first increase should be attributed to the facile RISC and suppressed non-radiative decay, the following reduction of luminescent intensity and lifetime is due to significantly suppressed RISC; as the temperature decreases, the T$_1$ (540 nm) emission gradually emerged and strengthened because of the combined effect of the fully suppressed nonradiative relaxation and RISC at low temperatures, showing an enhancement of lifetime (Fig. 3e). Based on the systematically experimental understanding, we concluded a possible mechanism including energy transfer and direct excitation processes that enable this spectacular hyperafterglow emission (Fig. 3f). For 285 nm excitation, photoexcited singlet excitons (S$_1^H$) in PAM can effectively transfer to the singlet state of MR-TADF guest VQA (S$_1^G$) followed by a fast ISC for generating triplet excitons of VQA (T$_1^G$); for 466 nm excitation, due to the lack of absorption, PAM cannot be excited and only isolated VQA was triggered; the photoexcited S$_1^G$ excitons can be promptly transferred to T$_1^G$; the boosted T$_1^G$ excitons through FRET (285 nm) and/or directly excited (466 nm) were usefully stabilized via the rigid hydrogen bond network of PAM; with facile RISC process, the stabilized and spin-forbidden T$_1^G$ excitons can transfer to spin-allowed S$_1^G$, thus conferring highly efficient hyperafterglow emission from isolated MR-TADF chromophore in the developed polymeric system. To demonstrate the vital role of the sensitizing process in enabling efficient hyperafterglow emission, a rigid polymer of PAAQA was also synthesized by replacing acrylamide with acrylic acid. As shown in Supplementary Fig. 35, although PAAQA demonstrated a comparable FWHM of delayed PL to that of PAMQA$_3$, the lifetime was ~6 folds lower than that of PAMQA$_3$ due to the lack of energy transfer from the polymer matrix to MR-TADF guest, suggesting the vital role of sensitizing process in enabling efficient hyperafterglow emission.

### Universality of design

To indicate the university of this sensitizing and stabilizing isolated fluorescence chromophores mechanism in exploiting hyperafterglow materials, other two copolymers with different afterglow sensitizers and/or emissive MR-TADF guests were constructed (Supplementary Section 1). First, to enlarge the lifetime of this sublime hyperafterglow, an afterglow host PAMCz with an ultralong lifetime of 4.2 s (Supplementary Fig. 36) was selected to replace PAM[17], and VQA was chosen as

a guest, a copolymer PAMCzQA was designed and synthesized (Fig. 4a). Because of the large spectral overlap between the delayed PL spectrum of PAMCz and the absorption spectrum of VQA, efficient energy transfer from PAMCz to VQA was maintained (Supplementary Fig. 37, top panel)[17]. Expectantly, a hyperafterglow emission peaked at 520 nm with a FWHM of 46 nm and CIE coordinate of (0.27, 0.67) was achieved in PAMCzQA (Fig. 4b, c). More importantly, time-resolved emission spectra excited by 285 nm demonstrated an increased ultralong (1.64 s) and stable hyperafterglow emission within increasing delayed time (Fig. 4d, e). To further modulate the hyperafterglow emission color, a red narrowband emitter of VQS was developed as guest[55] and the constructed PAMCzQA was used as afterglow host because its delayed PL spectrum is well overlapped with the absorption spectrum of VQS (Supplementary Fig. 37, bottom panel). Strikingly, a bright and stable red (636 nm) hyperafterglow emission with a lifetime of 1.16 s, FWHM of 56 nm and CIE coordinate of (0.66, 0.31) was achieved (Fig. 4b–e). Compared to the lifetimes of PAMCz (414 nm) and PAMCzQA (520 nm), obviously decreased lifetimes at these two emission bands were observed in PAMCzQAQS. This means that the FRET should dominate the energy transfer process in PAMCzQAQS (Supplementary Fig. 38 and Supplementary Table 4). Notably, after the copolymerization of VQS into PAMCzQA, the FRET efficiency further increased from 61.1% (PAMCzQA) to 72.2% (PAMCzQAQS).

### Hyperafterglow LEDs and displays

Benefiting from the distinguishable hyperafterglow emission, the potential afterglow lighting and display applications were explored as proof of concept[56]. The prototype afterglow lighting emitting diodes (LEDs) were developed (Fig. 5a) using the self-designed lampshade of red and green hyperafterglow polymer films and a UV LED chip ($\lambda_{ex}$ = 285 nm). As shown in Fig. 5b and Supplementary Fig. 39a, the hyperafterglow LED exhibited typical and stable steady-state electroluminescent (EL) and delayed EL features, showing a fixed emission peak at 504 and 636 nm as well as FWHM of ~40 and ~54 nm at different driving voltages and varied delay times for PAMQA$_3$ and PAMCzQAQS films, respectively. Moreover, low turn-on voltages of 3.0 and 3.1 V as well as maximum luminescence of 3023 and 1412 cd m$^{-2}$ were also realized in PAMQA$_3$ and PAMCzQAQS endowed green and red hyperafterglow LEDs, respectively (Fig. 5c and Supplementary Fig. 39b). In light of the excellent LED performance, an archetypal hyperafterglow display panel was further developed using PAMQA$_3$ film as an emissive layer. The transparent and uniform hyperafterglow panel showing remarkable afterglow intensity after removal of UV light irradiation can be facilely fabricated through the gradual evaporation of PAMQA$_3$ aqueous solution (Fig. 5d, e). Through conventional co-assembling of hyperafterglow film and circuitry-controlled LED array, the hyperafterglow display panel can be easily constructed (Supplementary Figs. 40–42). With the aid of mask technology, varied high-resolution afterglow patterns of "NJUPT" and "IAM" logos were readily accessible (Fig. 5f). Also, different afterglow digit numbers and paths can be conveniently regulated by modulating the circuitry-controlled LED array (Fig. 5g, Supplementary Movies 1, 2), opening up the possibility of constructing DC driven hyperafterglow display.

### Discussion

In summary, we have succussed in demonstrating a valuable strategy for realizing hyperafterglow with high luminescence intensity, ultralong lifetime, and narrowed emission band from isolated fluorescence MR chromophores. This strategy relies on simultaneously sensitizing and stabilizing isolated MR TADF emitters through afterglow polymer host for efficient FRET and RISC processes to transfer the spin-forbidden triplet excitons to the singlet state of MR-TADF guest for highly efficient hyperafterglow emission. Strikingly, multicolor hyperafterglow polymers with FWHM of ~40 nm, high PLQYs of ~88.9%, and an ultralong lifetime of 1.64 s were achieved in a set of

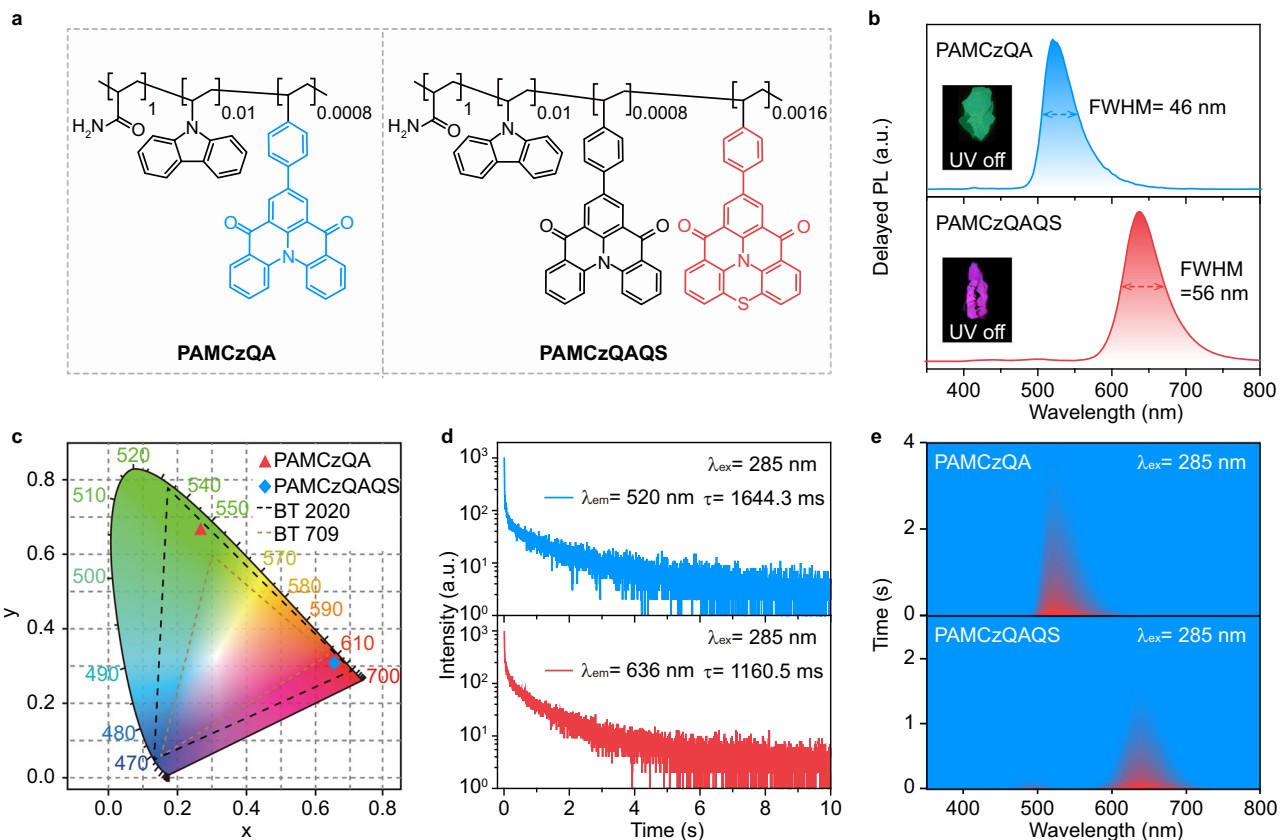

**Fig. 4 | Mechanism and feasibility confirmation of hyperafterglow emission.** **a** Molecule structures of PAMCzQA and PAMCzQAQS. **b** Delayed PL spectra (10 ms delay) of PAMCzQA (top panel) and PAMCzQAQS (bottom panel) films. Inset: photographs of PAMCzQA (top panel) and PAMCzQAQS (bottom panel) films taken upon turning off a 285 nm UV lamp. **c** CIE chromaticity diagram of the delayed PL emission of PAMCzQA and PAMCzQAQS films ($\lambda_{ex}$ = 285 nm). **d, e** Lifetime decay profiles (**d**) and transient emission decay images (**e**) of PAMCzQA (top panel) and PAMCzQAQS (bottom panel) films under 285 nm UV light excitation.

water-soluble copolymers. With this extraordinary hyperafterglow emission, the prototype applications of hyperafterglow LED and display panels were established. This work provides a design map for constructing hyperafterglow materials, shedding lighting on the discovery of advanced afterglow materials to overcome the inherent conflict between the ultralong lifetime and efficiency, which facilitates a myriad of possibilities for afterglow lighting and displaying.

## Methods

### Materials

All reagents, unless otherwise specified, were purchased from Energy Chemical, AOB Chem, and used without further purification. Manipulations involving air-sensitive reagents were performed in an atmosphere of dry argon (Ar).

### General procedure of radical polymerization

In an argon atmosphere, 0.01 equivalent (eq) of 2,2'-azobis(2-methyl-propionitrile) (AIBN) and 1.0 eq of vinyl derivative were dissolved in 25 mL freshly distilled tetrahydrofuran (THF). The mixture was heated to 55 °C for 16 h, during which the white, green, or red solid was constantly precipitated out from the solution. Then, the mixture was cooled to room temperature and added to methanol to precipitate polymeric materials, then the crude product was filtered, followed by washing with PE and DCM, acetone in sequence. Then the solid was dissolved in deionized water and dialyzed by a dialysis tube (molecular weight cut-off = 1000) for 72 h.

**PAMQA_1.** Following the general procedure of radical polymerization using VQA (15.96 mg, 0.04 mmol, 1.00 eq), acrylamide (3.55 g, 50.0 mmol, 1250 eq), and an appropriate amount of AIBN (82.07 mg, 0.5004 mmol, 12.51 eq) in 25 mL freshly distilled THF to afford 3.20 g green powder polymer with a yield of 89.7%. $M_n$ = 7404 Da; $M_w$ = 25512 Da; PDI = 3.44.

**PAMQA_2.** Following the general procedure of radical polymerization using VQA (3.99 mg, 0.01 mmol, 1.00 eq), acrylamide (3.55 g, 50.0 mmol, 5000 eq), and an appropriate amount of AIBN (82.02 mg, 0.5001 mmol, 50.51 eq) in 25 mL freshly distilled THF to afford 3.00 g green powder polymer with a yield of 84.4%. $M_n$ = 6651 Da; $M_w$ = 20261 Da; PDI = 3.04.

**PAMQA_3.** Following the general procedure of radical polymerization using VQA (2.00 mg, 0.005 mmol, 1.00 eq), acrylamide (3.55 g, 50.0 mmol, 10000 eq), and an appropriate amount of AIBN (82.01 mg, 0.50005 mmol, 100.01 eq) in 25 mL freshly distilled THF to afford 3.25 g green powder polymer with a yield of 91.5%. $M_n$ = 7990 Da; $M_w$ = 25268 Da; PDI = 3.16.

**PAMQA_4.** Following the general procedure of radical polymerization using VQA (0.20 mg, 0.0005 mmol, 1.00 eq), acrylamide (3.55 g, 50.0 mmol, 100,000 eq), and an appropriate amount of AIBN (82.00 mg, 0.500005 mmol, 1000.01 eq) in 25 mL freshly distilled THF to afford 3.10 g green powder polymer with a yield of 87.3%. $M_n$ = 7937 Da; $M_w$ = 24746 Da; PDI = 3.11.

**Polyacrylamide (PAM).** Following the general procedure of radical polymerization using acrylamide (3.55 g, 50.0 mmol, 100 eq), and an appropriate amount of AIBN (82.00 mg, 0.5 mmol, 1 eq) in 25 mL

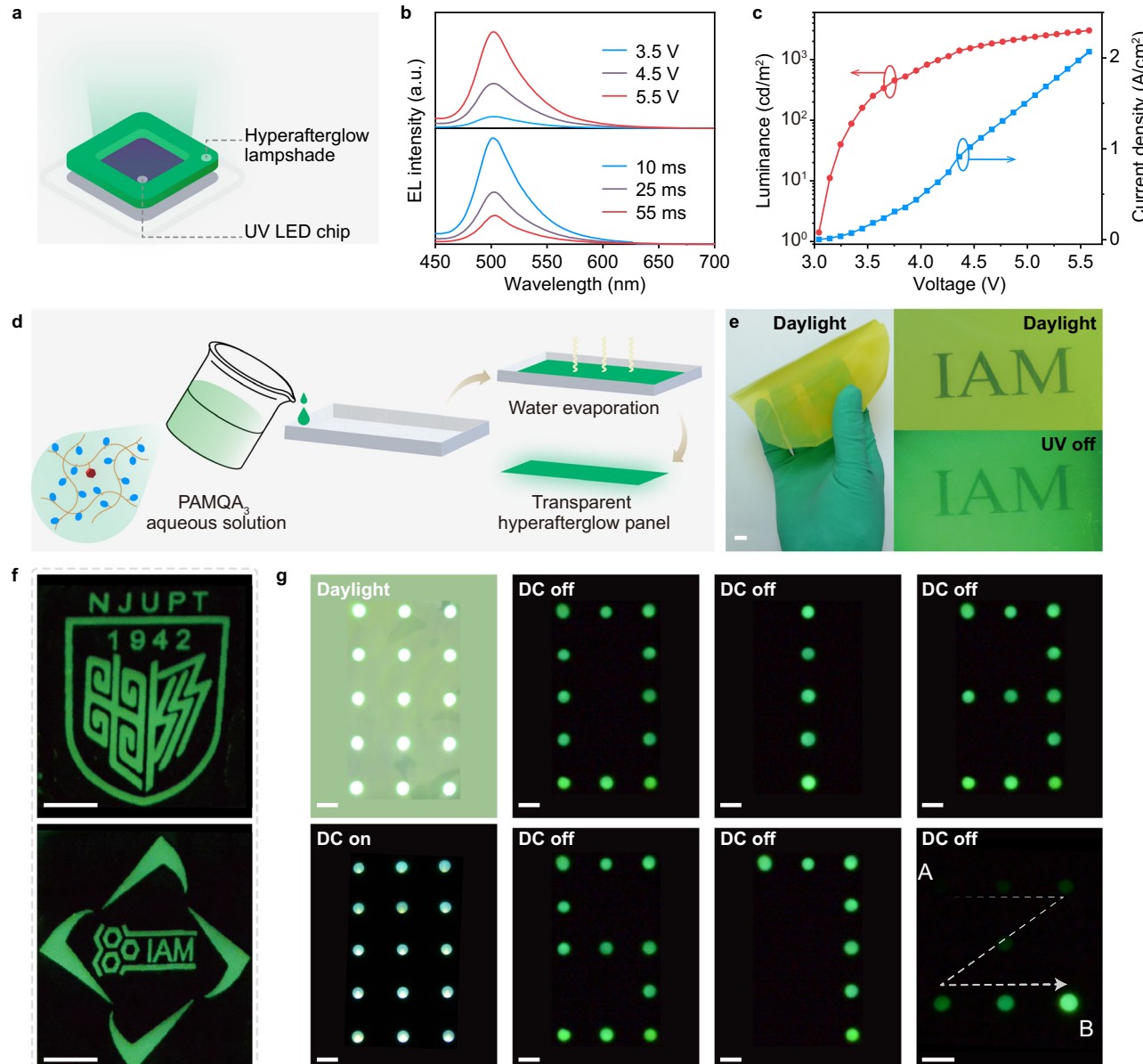

**Fig. 5 | Demonstration of hyperafterglow lighting and display. a** Schematic diagram of hyperafterglow LED. **b** SSEL (top panel) and delayed EL (bottom panel) spectra of hyperafterglow LED at varied driving voltages (top panel) and delayed times (bottom panel). **c** Current density–voltage–luminescence curves of hyperafterglow LED. **d** Fabrication of transparent hyperafterglow panel. **e** Photographs of the fabricated large area hyperafterglow panel under daylight and ceasing of UV light excitation. **f** Demonstration of hyperafterglow patterns via masked mask technology taken after the removal of 285 nm UV light. **g** Photograph of the hyperafterglow display panel and varied digital display items recorded under power supply on and off. The scale bars are 0.75 cm and the arrow indicates the afterglow path display from A to B.

freshly distilled THF to afford 3.30 g white powder polymer with a yield of 92.9%. $M_n$ = 8973 Da; $M_w$ = 27191 Da; PDI = 3.03.

**PAMCzQA.** Following the general procedure of radical polymerization using VQA (15.96 mg, 0.04 mmol, 1.00 eq), acrylamide (3.55 g, 50.0 mmol, 1250 eq), vinyl carbazole (96.5 mg, 0.5 mmol, 12.50 eq) and an appropriate amount of AIBN (82.90 mg, 0.5054 mmol, 12.635 eq) in 25 mL freshly distilled THF to afford 3.00 g green powder polymer with a yield of 81.9%. $M_n$ = 8284 Da; $M_w$ = 26698 Da; PDI = 3.22.

**PAMCzQAQS.** Following the general procedure of radical polymerization using VQA (15.96 mg, 0.04 mmol, 1.00 eq), VQS (34.33 mg, 0.08 mmol, 2.00 eq), acrylamide (3.55 g, 50.0 mmol, 1250 eq), vinyl carbazole (96.5 mg, 0.5 mmol, 12.50 eq) and an appropriate amount of AIBN (83.02 mg, 0.5062 mmol, 12.655 eq) in 25 mL freshly distilled THF to afford 2.8 g red powder polymer with a yield of 75.7%. $M_n$ = 7289 Da; $M_w$ = 15037 Da; PDI = 2.06.

**Photophysical measurements**
The steady-state UV absorption data were collected on a Jasco V-750 spectrophotometer. The SSPL spectra delayed PL spectra, and lifetimes of the organic afterglow were measured using an Edinburgh FLS980 fluorescence spectrophotometer equipped with a xenon arc lamp (Xe900) and a microsecond (μs) flash-lamp (uF900), respectively. For fluorescence decay measurements, a picosecond pulsed light-emitting diode (EPLED-295, wavelength: 300 nm, pulse width: 833.7 ps; VPL-375, wavelength: 375 nm, pulse width: 120 μs) was used.

## Data availability

The data that support the plots within this paper and other findings of this study are available from the corresponding author on request. Source data are provided with this paper.

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

## Acknowledgements

This study was supported in part by the National Natural Science Foundation of China (22075149 awarded to Y.T., 62075102 awarded to H.L. and 22275097 awarded to R.C.), the Jiangsu Specially-Appointed Professor Plan, the Six Talent Plan of Jiangsu Province (XCL-049 awarded to Y.T.), Hua Li Talents Program of Nanjing University of Posts and Telecommunications (awarded to Y.T.), Natural Science Fund for Colleges and Universities in Jiangsu Province (20KJB430001 awarded to H.L.), the Open Research Fund of Songshan Lake Materials Laboratory (2022SLABFN16 awarded to Y.T.), China Postdoctoral Science Foundation funded project (2018M642284 awarded to H.L.), and Nanjing University of Posts and Telecommunications Start-up Fund (NUPTSF) (NY219007 awarded to Y.T., NY220151 awarded to G.X. and NY217140 awarded to H.L.).

## Author contributions

X.Z., Y.T., R.C., and W.H. conceived the experiments and wrote the paper. X.Z., M.Z., C.Z., Z.G., and F.H. were primarily responsible for the experiments. X.Z, G.X., N.G., H.C., A.L., W.Z., Xin.Zhang., Hui.Li. Huanhuan.Li and Y.Z. measured and analyzed the photophysical properties. Yewen.Zhang. and P.L. performed the computational calculations. X.Z., M.Z., X.X., and F.H. fabricated the applications. All authors contributed to data analyses.

## Competing interests

The authors declare no competing interests.
