## [Peer Review File · Nature Communications]

Multicolor hyperafterglow from isolated fluorescence chromophoresReviewers' Comments:

Reviewer #1:

Remarks to the Author:

The paper entitled "Multicolor hyperafterglow from isolated fluorescence chromophores" presented a simple and efficient strategy to develop hyperafterglow polymers through the incorporation of MR-TADF chromophore into the afterglow polymer matrix. The designed polymers exhibited maximum photoluminescent efficiencies of 88.9%, minimum full-width at half-maximums (FWHMs) of 38 nm, and ultralong lifetimes of 1.64 s under ambient conditions. The photophysical and mechanism investigations have been systematically performed, and the manuscript is also well organized. Considering the novelty and broad interests of this work that would be useful for the development of high-performance hyperafterglow polymers, I recommend this work for publication in Nature Communications after a minor revision. Some comments are as follows:

1. Why the PAMQA_x showed slight larger FWHMs in SSPL spectra compared with delayed PL spectra in Figure 1a and Figure S9?
2. Why the efficiency of PAMQA₃ with 285 nm excitation is higher than that of 466 nm excitation in Figure 1c?
3. Besides PAM, can PAA be served as the matrix to boost the hyperafterglow emission from MR-TADF?
4. How about the FRET efficiency for PAMQA_x? What energy transfer process should be responsible for the PAMQA₃ since it has two-step energy transfer including PAM to QA and PAMQA to QS?
5. As shown in Figure 5, these applications do not show the potential of colorful hyperafterglow, which should be improved.

Reviewer #2:

Remarks to the Author:

In this manuscript, Zhang and co-workers reported an interesting strategy to construct multicolor hyperafterglow polymers through sensitizing and stabilizing isolated MR TADF emitter with the aid of rigid afterglow polymer host. Benefiting from the combined effect of intrinsic high color purity and efficiency of MR-TADF emitter and ultralong lifetime of polymer host, the high PLQY of 88.9%, small FWHMs of ~38 nm, and ultralong lifetimes up to 1.64 s were achieved in the developed polymers under ambient conditions. These results are helpful to further develop organic afterglow systems with high color purity and efficiency. Therefore, I think the paper is suitable to be published in Nature Communications after some minor revision. Detailed questions that need to be addressed before publication as shown below:

1. In this work, the important role of sensitizing and stabilizing has been demonstrated. To further prove it, the check experiment should be added. For example, the authors could replace PAM with PAA and/or PVP (Nanoscale. 2019, 11, 18311-18319. Light, Science & Applications. 2022, 11, 163) to construct hyperafterglow polymer.
2. The authors are suggested to study the hyperafterglow property when the MR-TADF emitter was physically doped into the corresponding polymer matrixes, such as PMMA, PVA, PVP, and PAM, etc.
3. The oscillator strength of afterglow (phosphorescence) is intrinsically small in purely organic polymer, and the energy transfer process from PAM to VQA emitter can not possibly be efficient. Because of the inherent low PLQY of non-conventional luminescent polymer of PAM host, how can the PLQY of hyperafterglow polymer be largely enhanced after FRET?
4. For the energy transfer process, the lifetime of energy acceptor should be lower than the energy donor, but the lifetimes of emission band at 504 nm (τ_G , VQA) were slightly larger than these of donor (τ_H , PAM) in PAMQA₁₋₃ (Table S2).
5. How did the authors prepare the large area hyperafterglow polymer film for display applications? It is essential difficulty to achieve a large and uniform film.
6. Why the PAMQA₃ film still had a lifetime over 100 ms at 504 nm under 466 nm excitation at 77 K, the RISC should be theoretically banned at this temperature.

7. Structural characterization should be provided, such as ^{13}C spectra and molecular weight test of VQS.
8. For the application as shown in Figure 5g, the white arrow in the path display was too small to identify.

Reviewer #3:

Remarks to the Author:

In this work, the authors demonstrated a simple yet effective strategy and achieved high-efficiency hyperafterglow through the copolymerization of MR-TADF monomers with acrylamide. In such single-component copolymer, the polyacrylamide chain segment is served as rigid host to sensitize MR-TADF emitters through triplet-to-singlet energy transfer. The photoluminescent efficiency and ultralong lifetimes of these hyperafterglow copolymers can reach up to 88.9% and 1.64 s, respectively. Specially, they exhibit FWHMs of around 40 nm. In addition, the authors applied the hyperafterglow copolymers to afterglow display applications. Therefore, this work can be published in Nature Communications after the following questions addressed.

1. The molar feed ratio of acrylamide (AM) and VQA in the research is very small (1:0.0008 for PAMQA1; 1: 0.0002 for PAMQA2; 1: 0.0001 for PAMQA3; 1: 0.00001 for PAMQA4). How about the reproducibility of these materials and their afterglow performance? The authors should provide evidence to verify the VQA monomer was truly covalently linked with AM.
2. It is very interesting that even at the extremely low molar feed ratio of 0.0001 (PAMQA3), the energy transfer from polyacrylamide chain segment to VQA is efficient. The authors should discuss more about the energy transfer process. Is the radiative energy transfer existed in the system?
3. In Fig. 2a, why the FWHM of delayed PL is smaller than SSPL?
4. The FWHM is a key parameter for these hyperafterglow copolymers, especially for afterglow emission. Can the author offer the FWHM values as a function of time for these materials? If the variation of FWHM at different delayed time exists, the authors should explain it.
5. For hyperafterglow display application, it looks like the green afterglow emission intensity is different in the LED array. Why some bulbs are very bright, but some are dim? The authors should describe more about the operating principle of circuitry-controlled LED array in the SI.

Reviewer comments:**Reviewer: #1**

Comment 1: *The paper entitled “Multicolor hyperafterglow from isolated fluorescence chromophores” presented a simple and efficient strategy to develop hyperafterglow polymers through the incorporation of MR-TADF chromophore into the afterglow polymer matrix. The designed polymers exhibited maximum photoluminescent efficiencies of 88.9%, minimum full-width at half-maximums (FWHMs) of 38 nm, and ultralong lifetimes of 1.64 s under ambient conditions. The photophysical and mechanism investigations have been systematically performed, and the manuscript is also well organized. Considering the novelty and broad interests of this work that would be useful for the development of high-performance hyperafterglow polymers, I recommend this work for publication in Nature Communications after a minor revision. Some comments are as follows:*

Author reply: Thanks for the referee’s professional comments and kind recommendation of our work. We have carefully addressed the points raised by the referee.

(1) Why the PAMQA_x showed slight larger FWHMs in SSPL spectra compared with delayed PL spectra in Figure 1a and Figure S9?

Author reply: We appreciate the professional question raised by the referee. When a TADF material becomes excited, it exhibits a prompt fluorescence and a delayed fluorescence of similar wavelength in SSPL spectra. The delayed fluorescence dominates the emission process in the delayed PL spectra. Therefore, the mixed emission species from prompted fluorescence and delayed fluorescence should have slightly larger FWHM in SSPL spectra compared to the delayed PL spectra with only the delayed fluorescence. This is a common phenomenon for organic TADF-type afterglow materials (*Sci. Adv.* **2017**, *3*, e1603171; *Chem. Commun.* **2022**, *58*, 11418; *Adv. Funct. Mater.* **2022**, *32*, 2110207). We have involved the explanation in the revised manuscript. Thanks a lot.

Added text:

Page 2 of the revised manuscript:

Compared to the delayed PL spectra, the mixed emission species from prompted fluorescence and delayed fluorescence leads to the slightly larger FWHM in SSPL spectra¹³.

(2) Why the efficiency of PAMQA₁ with 285 nm excitation is higher than that of 466 nm excitation in Figure 2c?

Author reply: Many thanks for the careful review and thoughtful question. After double checked the experimental results, we find that we make a mistake when we draw Fig 2c. The 466 nm excited PLQY (68.86%) of PAMQA₁ is actually higher than that of 285 nm excitation (66.57%) (Fig 2c and Table R1). We have updated Fig 2c in the revised manuscript. Thanks again.

Updated Figure:

Page 3 of the revised manuscript:

Fig. 2 | Photophysical Properties of hyperafterglow polymers under ambient conditions. a Normalized steady-state PL (SSPL) and delayed PL (10 ms delay) spectra (red line) of PAMQA₃ film ($\lambda_{\text{ex}} = 285$ nm). Inset: photographs of PAMQA₃ film taken upon turning on (top panel) and off (bottom panel) a 285 nm UV lamp. **b** CIE chromaticity diagram for SSPL and delayed PL emission of PAMQA₃ ($\lambda_{\text{ex}} = 285$ nm). **c** PLQYs of PAMQA_x films upon 285 nm and 466 nm excitation. **d** Lifetime decay profiles of emission band at 504 nm of PAMQA₃ film upon 285 nm and 466 nm excitation. **e** Transient emission decay images of PAMQA₃ film ($\lambda_{\text{ex}} = 285$ nm). **f** Normalized SSPL spectra of PAMQA₃ aqueous solution (top panel) and film (bottom panel). **g** 2D-WAXS pattern of PAMQA₃ film. The inset shows a schematic illustration of the isolated afterglow from a single-component copolymer. **h, i** Excitation-SSPL (**h**) and excitation-delayed PL (**i**) (25 ms delay) mappings of PAMQA₃ film.

Table R1. PLQYs of PAMQA_x films upon 285 nm and 466 nm excitation.

Polymer	PLQY ($\lambda_{\text{ex}}=285$ nm)	PLQY ($\lambda_{\text{ex}}=466$ nm)
PAMQA ₁	66.57	68.86
PAMQA ₂	65.30	72.55
PAMQA ₃	63.32	74.85
PAMQA ₄	53.01	88.86

(3) Besides PAM, can PAA be served as the matrix to boost the hyperafterglow emission from MR-TADF?

Author reply: We appreciate the constructive suggestion raised by the referee. As suggested by the referee, we have performed the control experiments by copolymerization of acrylic acid (AA) (PAAQA) and MR-TADF VQA with a feed ratio of 1:0.0001. PAAQA exhibits a narrow emission with FWHM of ~44 nm from MR-TADF VQA and ~8 ms lifetime when excited by 285 nm UV light and/ or 466 nm visible light (Supplementary Fig. 35). Although PAAQA demonstrated a comparable FWHM of delayed PL to that of PAMQA₃, the lifetime was ~6 folds lower than that of PAMQA₃, probably due to the lack of energy transfer from polymer matrix to MR-TADF guest (VQA). Because of the short lifetime, *no obvious afterglow emission* could be found in PAAQA. Considering the above experimental results, PAA is not a good matrix to boost the hyperafterglow emission. In the revised manuscript, we have involved these experimental results to demonstrate the vital role of PAM in enabling efficient hyperafterglow emission. Many thanks.

Added text and figure:

Page 5-6 of the revised manuscript and Page S33 of the revised Supplementary Information:

To demonstrate the vital role of sensitizing process in enabling efficient hyperafterglow emission, a rigid polymer of PAAQA was also synthesized by replacing acrylamide with acrylic acid. As shown in Supplementary Fig. 35, although PAAQA demonstrated a comparable FWHM of delayed PL to that of PAMQA₃, the lifetime was ~6 folds lower than that of PAMQA₃ due to the lack of energy transfer from polymer matrix to MR-TADF guest, suggesting the vital role of sensitizing process in enabling efficient hyperafterglow emission.

Figure S35. SSPL and delayed PL spectra (10 ms delay) of PAAQA film upon (a) 285 nm UV light and (b) 466 nm visible light excitation. (c) Lifetime decay profiles of emission band at 508 nm of PAAQA film upon 285 nm UV light and 466 nm visible light excitation.

(4.1) How about the FRET efficiency for PAMQA_x?

Author reply: Thanks for the referee's professional question. PAMQA_x exhibit high energy transfer efficiencies of up to 94.7% for PAMQA_x (Supplementary Table 3) as calculated from the amplitude averaged lifetimes (438 nm) of PAM and PAMQA_x. These FRET efficiencies have been added in the revised manuscript.

Added text and figure:

Page 5 of the revised manuscript and Page S29 of the revised Supplementary Information:

PAMQA_x exhibit high energy transfer efficiencies of up to 94.7% (Supplementary Table 3),

as calculated from the amplitude averaged lifetimes (438 nm) of PAM and PAMQA_x (Supplementary Fig. 31).

Figure S31. Lifetime decay profiles of PAMQA_x and PAM films upon 285 nm UV light excitation.

Table S3. Phosphorescence amplitude lifetime and corresponding energy transfer efficiency of PAMQA_x.

Polymer	λ_p^H (nm)	$\tau_{amp}^{H,P}$ (ms)	λ_p^G (nm)	$\tau_{amp}^{G,P}$ (ms)	Φ_{P-FRET} (%)
PAM	438	170	--	--	--
PAMQA ₁	438	9	524	15	94.7
PAMQA ₂	438	11	514	21	93.5
PAMQA ₃	438	20	504	30	88.2
PAMQA ₄	438	23	496	34	86.5

(4.2) What energy transfer process should be responsible for the PAMCzQAQS since it has two-step energy transfer including PAMCz to QA and PAMCzQA to QS?

Author reply: Thanks for the referee's professional question. According to the analyses of lifetime, the FRET should be responsible for the PAMCzQAQS. As shown in Supplementary Fig. 38 and Supplementary Table 4, obvious decreased lifetime (414 nm) was observed in PAMCzQA in comparison with that in PAMCz, suggesting that the FRET should be responsible for energy transfer from PAMCz to VQA. After the copolymerization of VQS into PAMCzQA (PAMCzQAQS), the lifetime (520 nm) of PAMCzQA was decreased from 0.7 to 0.2 s, suggesting the facilitated FRET from PAMCzQA to VQS. Besides, the lifetime of host PAMCz (414 nm) further reduced from 1.4 to 1.0 s when VQS was incorporated into PAMCzQA, demonstrating also that copolymerization of VQS into PAMCzQA can further enhance the FRET process. According to the fitted amplitude lifetime, the FRET efficiencies were calculated to be 61.1% to 72.2% for PAMCzQA and PAMCzQAQS, respectively. Many thanks.

Added text and figure:

Page 7 of the revised manuscript and Page S35 of the revised Supplementary Information:

Compared to the lifetimes of PAMCz (414 nm) and PAMCzQA (520 nm), obvious decreased lifetime at these two emission bands were observed in PAMCzQAQS. This means that the FRET should dominate the energy transfer process in PAMCzQAQS (Supplementary Fig. 38 and Supplementary Table 4). Notably, after the copolymerization of VQS into PAMCzQA, the FRET efficiency further increased from 61.1% (PAMCzQA) to 72.2% (PAMCzQAQS).

Figure S38. Lifetime decay profiles of PAMCz, PAMCzQA and PAMCzQAQS at the emission bands of (a) 414 and (b) 520 nm upon 285 nm UV light excitation.

Table S4. Phosphorescence amplitude lifetime and corresponding energy transfer efficiency of PAMCzQA and PAMCzQAQS.

Polymer	λ_p^H (nm)	$\tau_{amp}^{H,P}$ (s)	λ_p^{VQA} (nm)	τ_{amp}^{VQA} (s)	Φ_{P-FRET} (%)
PAMCz	414	3.6	—	—	—
PAMCzQA	414	1.4	520	0.7	61.1
PAMCzQAQS	414	1.0	520	0.2	72.2

(5) As shown in Figure 5, these applications do not show the potential of colorful hyperafterglow, which should be improved.

Author reply: We appreciate the very nice suggestions. To demonstrate the potential of colorful hyperafterglow, we selected PAMCzQAQS as lampshade for the construction of red hyperafterglow LEDs. As shown in Supplementary Fig. 39, the red hyperafterglow LED exhibits gradually increased afterglow intensities at increased the driving voltages for stable hyperafterglow emission with elongated delayed time. We have added these results in the

revised manuscript and Supplementary Materials. Many thanks.

Updated text and added figures:

Page 7 of the revised manuscript and Page S37 of the revised Supplementary Information:

The prototype afterglow lighting emitting diodes (LEDs) were developed (Fig. 5a) using the self-designed lampshade of red and green hyperafterglow polymer films and a UV LED chip ($\lambda_{\text{ex}} = 285 \text{ nm}$). As shown in Fig. 5b and Supplementary Fig. 39a, the hyperafterglow LED exhibited typical and stable steady-state electroluminescent (EL) and delayed EL features, showing a fixed emission peak at 504 nm and 636 nm as well as FWHM of $\sim 40 \text{ nm}$ and $\sim 54 \text{ nm}$ at different driving voltages and varied delay times for PAMQA₃ and PAMCzQAQS films, respectively. Moreover, low turn-on voltages of 3.0 and 3.1 V as well as maximum luminescence of 3023 and 1412 cd m^{-2} were also realized in PAMQA₃ and PAMCzQAQS endowed green and red hyperafterglow LEDs, respectively (Fig. 5c and Supplementary Fig. 39b).

Figure S39. (a) SSEL (top panel) and delayed EL (bottom panel) spectra of red hyperafterglow LED at varied driving voltages (top panel) and delayed times (bottom panel). (b) Current density-voltage-luminescence curves of red hyperafterglow LED.

Reviewer: #2

Comment2: *In this manuscript, Zhang and co-workers reported an interesting strategy to construct multicolor hyperafterglow polymers through sensitizing and stabilizing isolated MR TADF emitter with the aid of rigid afterglow polymer host. Benefiting from the combined effect of intrinsic high color purity and efficiency of MR-TADF emitter and ultralong lifetime of polymer host, the high PLQY of 88.9%, small FWHMs of ~38 nm, and ultralong lifetimes up to 1.64 s were achieved in the developed polymers under ambient conditions. These results are helpful to further develop organic afterglow systems with high color purity and efficiency. Therefore, I think the paper is suitable to be published in Nature Communications after some minor revision. Detailed questions that need to be addressed before publication as shown below:*

Author reply: Thanks for the professional comments and recommendation of our work. We have revised the manuscript according to the referee's suggestions.

(1) In this work, the important role of sensitizing and stabilizing has been demonstrated. To further prove it, the check experiment should be added. For example, the authors could replace PAM with PAA and/or PVP (Nanoscale. 2019, 11, 18311-18319. Light, Science & Applications. 2022, 11, 163) to construct hyperafterglow polymer.

Author reply: We appreciate the point raised by the referee. As suggested by the referee, we have performed the control experiments by copolymerization of acrylic acid (**AA**) (**PAAQA**) or N-vinyl-2-pyrrolidone (**NVP**) (**PVPQA**) with MR-TADF **VQA** with a feed ratio of 1:0.0001. **PAAQA** and **PVPQA** exhibited narrow emission with FWHM of ~44 nm from MR-TADF **VQA** as well as ~8 ms and ~2 ms lifetime when excited by 285 nm UV light or 466 nm visible light (**Supplementary Fig. 35 and Figure R1**). Although **PAAQA** and **PVPQA** demonstrated comparable FWHMs of delayed PL to that of **PAMQA₃**, the lifetime was ~6 folds (**PAAQA**) and ~24 folds (**PVPQA**) lower than that of **PAMQA₃** due to the lack of sensitizing process through energy transfer from polymer matrix to MR-TADF guest (**VQA**). Because of the short lifetime, **no obvious afterglow emission** could be found in **PAAQA** and **PVPQA**. Considering the above experimental results, **PAA** and **PVP** are not good matrix to boost the hyperafterglow emission. In the revised manuscript, we have involved these experimental results and cited the related references to demonstrate the vital roles of both sensitizing and stabilizing processes in enabling efficient hyperafterglow emission. Many thanks.

Figure R1. (a) SSPL and delayed PL spectra (10 ms delay) of **PVPQA** film upon 285 nm UV light excitation. (b) Lifetime decay profiles of emission band at 480 nm of **PVPQA** film upon 285 nm UV light and 466 nm visible light excitation.

Added text and figures:

Page 5-6 of the revised manuscript and Page S33 of the revised Supplementary Information:

To demonstrate the vital role of sensitizing process in enabling efficient hyperafterglow emission, a rigid polymer of **PAAQA** was also synthesized by replacing acrylamide with acrylic acid. As shown in **Supplementary Fig. 35**, although **PAAQA** demonstrated a comparable FWHM of delayed PL to that of **PAMQA₃**, the lifetime was ~6 folds lower than that of **PAMQA₃** due to the lack of energy transfer from polymer matrix to MR-TADF guest, suggesting the vital role of sensitizing process in enabling efficient hyperafterglow emission.

Figure S35. SSPL and delayed PL spectra (10 ms delay) of PAAQA film upon (a) 285 nm UV light and (b) 466 nm visible light excitation. (c) Lifetime decay profiles of emission band at 508 nm of PAAQA film upon 285 nm UV light and 466 nm visible light excitation.

(2) The authors are suggested to study the hyperafterglow property when the MR-TADF emitter was physically doped into the corresponding polymer matrixes, such as PMMA, PVA, PVP, and PAM, etc.

Author reply: We thank the very constructive suggestion. According to the referee's suggestions, we have performed the comparative experiments using the direct physical

mixture by doping MR-TADF emitter of **VQA** into **PMMA**, **PVA**, **PVP** and **PAM** at the weight concentration of 0.1 wt.%, respectively. The physically blended polymer matrix and **VQA** systems demonstrate comparable FWHMs of delayed PL to that of **PAMQA₃**, but the lifetimes are up to ~70 folds (**VQA@PVP**) lower than that of **PAMQA₃** due to the lack of efficient energy transfer from polymer matrix to **VQA** and the poor intermixing between water-soluble polymer matrix and oil-soluble **VQA**. In the revised manuscript, we have involved these experimental results to prove the importance of copolymerization in achieving efficient hyperafterglow emission. Many thanks.

Figure R2. (a) Delayed PL spectra (5 ms delay) and (b) lifetime decay profiles of **VQA@PMMA**, **VQA@PVA**, **VQA@PVP** and **VQA@PAM** film excited by 285 nm UV light.

Added text and figures:

Page 3 of the revised manuscript and Page S23 of the revised Supplementary Information:

Notably, the copolymerization is much more effective than the physically blended polymer system of **VQA** and **PAM** to achieve ultralong lifetime of hyperafterglow emission (Supplementary Fig. 21).

Figure S21. (a) SSPL and delayed PL spectra (10 ms delay) and (b) lifetime decay profiles of VQA@PAM film excited by 285 nm UV light.

(3) The oscillator strength of afterglow (phosphorescence) is intrinsically small in purely organic polymer, and the energy transfer process from PAM to VQA emitter can not possibly be efficient. Because of the inherent low PLQY of non-conventional luminescent polymer of PAM host, how can the PLQY of hyperafterglow polymer be largely enhanced after FRET?

Author reply: We thank the referee for the interesting questions. The doping of highly efficient fluorescent acceptors (guests) into donors (hosts) could greatly increase the quantum yield of the composites due to the fast radiative decay of the acceptor (guest), which can effectively capture the exciton of donor (host) through energy transfer to reduce the non-radiative decay of donors. Therefore, the improvement of quantum yields after FRET process is possible (*J. Fluoresc.* **2002**, *12*, 97). In our work, the selected VQA guest has a high PLQY up to ~90.0% (*ACS Appl. Mater. Interfaces* **2019**, *11*, 13472; *Angew. Chem. Int. Ed.* **2022**, *61*, e202213697), whereas the steady-state photoluminescence of host PAM is relatively weak owing to the non-radiative decay of the excitons. By taking advantage of efficient energy transfer from PAM to VQA and highly efficient radiative transitions of excited excitons of the VQA, the non-radiative depopulation of excitons of PAM is significantly decreased, leading to the increased quantum yields of the resulting hyperafterglow polymers. In fact, the improvement of quantum yields after FRET process is widely observed in host-guest systems in previous works (**Table R2**).

Table R2. Improved quantum yields of photoluminescence after FRET process in previous works.

Reference	Quantum yields	Quantum yields after FRET
-----------	----------------	---------------------------

Nat. Commun. 2014, 5, 4016.	57%	88%
Chem. Eur. J. 2017, 24, 1151.	33%	37%
Adv. Mater. 2018, 30, 1800365.	13%	46%
Angew. Chem. Int. Ed. 2020, 59, 9393.	25.58%	53.76%
Nat. Photonics. 2021, 15, 203.	86%	97%

(4) For the energy transfer process, the lifetime of energy acceptor should be lower than the energy donor, but the lifetimes of emission band at 504 nm (τ_G , VQA) were slightly larger than these of donor (τ_H , PAM) in PAMQA₁₋₃ (Table S2).

Author reply: We thank the referee for the careful review and constructive questions. According to the previous reports (ACS Energy Lett. 2022, 7, 10) and our understandings, the larger lifetime of the VQA (τ_G) guest than that of the PAM host should be due to the fact that delayed PL emission of MR-TADF through the reversed intersystem crossing process will slightly elongate the lifetime of VQA guest. We hope our explanation can clarify the referee's questions.

(5) How did the authors prepare the large area hyperafterglow polymer film for display applications? It is essential difficulty to achieve a large and uniform film.

Author reply: We apologize for forgetting to provide the detailed preparation procedure. We have involved the specific preparation process in the revised Supplementary Information. Thanks a lot.

Added text:

Page S36 of the revised Supplementary Information:

Detailed procedure for the fabrication of hyperafterglow display panel: 5 g PAMQA₃ powders were dissolved in 15 mL deionized water, followed by the sonication for 30 mins under ambient conditions. Subsequently, the mixture was vigorously stirred at 60°C for 1 hour to obtain the transparent solution. Last, the well-mixed solution was poured into a clean Teflon box, followed by the natural evaporation of water. Because of the excellent film-forming ability inherited from PAM, the large and uniform polymer film could be easily achieved for display applications.

(6) Why the PAMQA₃ film still had a lifetime over 100 ms at 504 nm under 466 nm excitation at 77

K, the RISC should be theoretically banned at this temperature.

Author reply: Thank the referee for the professional and constructive question. As shown in **Figures 3c, d**, the emission from 504 nm can still be observed in SSPL and delayed PL spectra at 77 K but with decreased intensities compared to that at 298 K, which means that this RISC process is largely suppressed rather than totally prohibited. Therefore, the lifetime at the emission band of 504 nm was still larger than 100 ms due to the combined effect of the suppressed non-radiative decay and RISC process. Notably, this is a common phenomenon for organic TADF-type afterglow materials (*Angew. Chem. Int. Ed.* **2020**, *59*, 10032; *Adv. Mater.* **2020**, *32*, 2000936; *Sci. Adv.* **2017**, *3*, e1603171). We hope this explanation can clarify the referee's question. Thanks a lot.

(7) Structural characterization should be provided, such as ¹³C spectra and molecular weight test of VQS.

Author reply: We thank the referee for the useful suggestion. We have carried out ¹³C NMR and molecular weight measurements of **VQS**.

Added text and Figure:

Page S6 and S8 of the revised Supplementary Information:

¹³C NMR (101 MHz, 10% TFA-d in CDCl₃) δ=177.94, 140.96, 138.56, 136.88, 136.36, 136.04, 135.68, 135.29, 133.32, 132.05, 127.30, 126.88, 126.28, 123.52, 122.07, 121.11, 115.34. MALDI-TOF: m/z calcd for C₂₈H₁₅NO₂S [M]⁺: 429.080; Found: 429.262.

Figure S5. ^{13}C NMR spectrum of VQS in 10% TFA-d in CDCl_3

(8) For the application as shown in Figure 5g, the white arrow in the path display was too small to identify.

Author reply: We thank the referee for the kind guidance in treating figures. We have revised Figure 5g with an enlarged white arrow. Please see the updated Figure 5g in the revised manuscript.

Updated figure:

Page 6 of the revised manuscript:

Fig. 5 | Demonstration of hyperafterglow lighting and display. **a** Schematic diagram of hyperafterglow LED. **b** SSEL (top panel) and delayed EL (bottom panel) spectra of hyperafterglow LED at varied driving voltages (top panel) and delayed times (bottom panel). **c** Current density-voltage-luminescence curves of hyperafterglow LED. **d** Fabrication of transparent hyperafterglow panel. **e** Photographs of the fabricated large area hyperafterglow panel under daylight and ceasing of UV light excitation. **f** Demonstration of hyperafterglow patterns via masked mask technology taken after the removal of 285 nm UV light. **g** Photograph of the hyperafterglow display panel and varied digital display items recorded under power supply on and off. The scale bars are 0.75 cm.

Reviewer: #3

Comment3: *In this work, the authors demonstrated a simple yet effective strategy and achieved high-efficiency hyperafterglow through the copolymerization of MR-TADF monomers with acrylamide. In such single-component copolymer, the polyacrylamide chain segment is served as rigid host to sensitize MR-TADF emitters through triplet-to-singlet energy transfer. The photoluminescent efficiency and ultralong lifetimes of these hyperafterglow copolymers can reach up to 88.9% and 1.64 s, respectively. Specially, they exhibit FWHMs of around 40 nm. In addition, the authors applied the hyperafterglow copolymers to afterglow display applications. Therefore, this work can be published in Nature Communications after the following questions addressed.*

Author reply: Thanks for the professional comments and recommendation of our work. We have carefully addressed the points raised by the referee.

(1) The molar fed ratio of acrylamide (AM) and VQA in the research is very small (1:0.0008 for PAMQA₁; 1: 0.0002 for PAMQA₂; 1: 0.0001 for PAMQA₃; 1: 0.00001 for PAMQA₄). How about the reproducibility of these materials and their afterglow performance? The authors should provide evidence to verify the VQA monomer was truly covalently linked with AM.

Author reply: We thank the referee for the professional questions and suggestions. To demonstrate the reproducibility of our hyperafterglow polymer materials, we prepared **PAMQA₃** and tested their photophysical properties again. As shown in **Figure R2**, the hyperafterglow demonstrates minimal batch-to-batch variations in different batches of polymers, demonstrating the excellent reproducibility of the hyperafterglow emission. Moreover, to verify that **VQA** monomer was indeed embedded in the polymer through covalent linkage, we have performed the NMR measurements of the physically blended **PAM** and **VQA** system using the same mass ratio to that of the copolymer **PAMQA_x**. Compared to **PAMQA_x**, the chemical shifts of unsaturated double bonds (5-6 ppm) from the **VQA** monomer can be clearly observed in the physically blended **PAM** with **VQA** systems (**Supplementary Fig. 7**), but these chemical shift signals of the unsaturated double bonds are absent in the copolymers. These results suggest that the **VQA** should have indeed participated into the copolymer. Thanks again.

Figure R2. (a, b) SSPL and delayed PL spectra (10 ms delay) and (c, d) lifetime decay profile of PAMQA₃ film upon 285 nm UV light excitation.

Added text and Figures:

Page 2 of the revised manuscript and Page S12 of the revised Supplementary Information:

Compared to the physically mixed PAM and VQA (Supplementary Fig. 7) showing obvious chemical shifts from VQA monomer, the ¹H NMR spectra of PAMQA_x confirm that VQA should have indeed participated in the copolymers.

Figure S7. ¹H spectra of physically blended PAM and VQA in mixed D₂O and d-THF solution with same mass feed ratios to that of PAMQA_x.

(2) It is very interesting that even at the extremely low molar feed ratio of 0.0001 (PAMQA₃), the energy transfer from polyacrylamide chain segment to VQA is efficient. The authors should discuss

more about the energy transfer process. Is the radiative energy transfer existed in the system?

Author reply: We really appreciate the professional and thoughtful comments. As suggested by the referee, we have re-measured the lifetime (438 nm) of **PAMQA_x** and calculated the FRET efficiencies using the amplitude averaged lifetime. At the increased molar feed ratio of **VQA**, the afterglow emission and lifetime of **PAM** were gradually decreased, suggesting that the radiative energy transfer could be excluded. And, **PAMQA_x** exhibit high energy transfer efficiencies of up to 94.7% (**Supplementary Table 3**) calculated from the amplitude averaged lifetimes (438 nm) of **PAM** and **PAMQA_x** (**Supplementary Fig. 31**). Because of the high FRET efficiencies, the radiative energy transfer could be excluded. These results have been updated in the revised manuscript and Supplementary Materials (**Supplementary Fig. 31** and **Supplementary Table 3**). Many thanks.

Added text and figure:

Page 5 of the revised manuscript and Page S29 of the revised Supplementary Information:

PAMQA_x exhibit high energy transfer efficiencies of up to 94.7% (**Supplementary Table 3**) calculated from the amplitude averaged lifetimes (438 nm) of **PAM** and **PAMQA_x** (**Supplementary Fig. 31**).

Figure S31. Lifetime decay profiles of **PAMQA_x** and **PAM** films upon 285 nm UV light excitation.

Table S3. Phosphorescence amplitude lifetime and corresponding energy transfer efficiency of **PAMQA_x**.

Polymer	λ_p^H (nm)	$\tau_{amp}^{H,P}$ (ms)	λ_p^G (nm)	$\tau_{amp}^{G,P}$ (ms)	Φ_{P-FRET} (%)
PAM	438	170	--	--	--
PAMQA_x	438	9	524	15	94.7

PAMQA ₂	438	11	514	21	93.5
PAMQA ₃	438	20	504	30	88.2
PAMQA ₄	438	23	496	34	86.5

(3) In Fig. 2a, why the FWHM of delayed PL is smaller than SSPL?

Author reply: We appreciate the professional question raised by the referee. When a TADF material becomes excited, it exhibits a prompt fluorescence and then a delayed fluorescence of similar wavelength in SSPL. Compared to the SSPL spectra, only delayed fluorescence dominates the emission process in the delayed PL spectra, thus leading to a slightly narrower FWHMs. Notably, this is a common phenomenon for organic TADF-type afterglow materials (*Sci. Adv.* **2017**, *3*, e1603171; *Chem. Commun.* **2022**, *58*, 11418; *Adv. Funct. Mater.* **2022**, *32*, 2110207). We have involved these discussions in the revised manuscript. Thanks a lot.

Added text:

Page 2 of the revised manuscript:

Compared to the delayed PL spectra, the mixed emission species from prompted fluorescence and delayed fluorescence leads to the slightly larger FWHM in SSPL spectra¹³.

(4) The FWHM is a key parameter for these hyperafterglow copolymers, especially for afterglow emission. Can the author offer the FWHM values as a function of time for these materials? If the variation of FWHM at different delayed time exists, the authors should explain it.

Author reply: Many thanks for the thoughtful comments and questions. The FWHM values as a function of delayed time for PAMQA₃ was provided (**Supplementary Fig. 15**). With elongating the delayed time, the FWHM keeps almost constant showing ~ 3 nm variation.

Updated text and added Figures:

Page 3 of the revised manuscript and Page S21 of the revised Supplementary Information:

Time-resolved emission spectra demonstrated a long-lived and stable hyperafterglow emission showing ~ 3 nm FWHMs variation within the increasing delayed time (**Supplementary Fig. 15**).

Figure S15. (a) Delayed PL spectra of PAMQA₃ film with different delayed times. **(b)**The afterglow FWHM of PAMQA₃ film with different delayed time.

(5) For hyperafterglow display application, it looks like the green afterglow emission intensity is different in the LED array. Why some bulbs are very bright, but some are dim? The authors should describe more about the operating principle of circuitry-controlled LED array in the SI.

Author reply: We really appreciate the points raised by the referee. Because of the different height of the LED array, the intensities of bulbs are different. We have carefully adjusted the height of the LED array and retaken the afterglow photos. These photos showing uniform afterglow emission were updated in the revised Fig. 5g. Moreover, the operation principle of circuitry-controlled LED array was also updated in the revised Supplementary Information.

Added text and updated Figures:

Page 6 of the revised manuscript:

Fig. 5 | Demonstration of hyperafterglow lighting and display. **a** Schematic diagram of hyperafterglow LED. **b** SSEL (top panel) and delayed EL (bottom panel) spectra of hyperafterglow LED at varied driving voltages (top panel) and delayed times (bottom panel). **c** Current density-voltage-luminescence curves of hyperafterglow LED. **d** Fabrication of transparent hyperafterglow panel. **e** Photographs of the fabricated large area hyperafterglow panel under daylight and ceasing of UV light excitation. **f** Demonstration of hyperafterglow patterns *via* masked mask technology taken after the removal of 285 nm UV light. **g** Photograph of the hyperafterglow display panel and varied digital display items recorded under power supply on and off. The scale bars are 0.75 cm.

Page S36 and S38 of the revised Supplementary Information

The operation principle of circuitry-controlled LED array: Firstly, we assembled an UV LED array into the acrylic template (**Supplementary Figs. 40-41**), then the UV LED array was connected and controlled by a commercial programmable LED controller and transformer. Through programming controller, varied digit numbers and paths for display applications can be conveniently achieved by modulating the circuitry-controlled LED array. For example, by

manipulating the column of the control table, the digital numbers 0-9 can be easily achieved, and then by programming the row of the control table, the drive duration and interval time for digital numbers 0-9 can be precisely modulated. When the No.1-No.6 and No.10-No.15 LEDs of the column are red (Supplementary Fig. 42), the digital number 0 is lighting and continuous lights for 6 seconds (time value 1), then the LED array extinguishes for 6 seconds, and the next digit number will light for another 6 seconds (time value 1).

Figure S42. Schematic drawing of LED array (left) and the corresponding program for modulating the circuitry-controlled LED array to achieve the digital numbers of 0-9.

A list of changes made:

- (1) PLQY of **PAMQA_x** in **Fig. 2c** was updated.
- (2) Photographs for hyperafterglow display in **Fig. 5g** was updated.
- (3) ¹³C NMR spectra and molecular weight of **VQS** were measured and added in Supplementary Information (**Supplementary Fig. 5**).
- (4) ¹H spectra of physically blended **PAM** and **VQA** were measured and added in **Supplementary Fig. 7**
- (5) Delayed PL spectrum of **PAMQA₃** film with different delayed time and the afterglow FWHM of **PAMQA₃** film with different delayed time were added in **Supplementary Fig. 15**.
- (6) SSPL and delayed PL spectra, and lifetime decay profile of **VQA@PAM** were measured in **Supplementary Fig. 21**.
- (7) Lifetime decay profiles and relevant parameter of **PAMQA_x** and **PAM** at 438 nm were updated in Supplementary Information (**Supplementary Fig. 31 and Supplementary Table 3**).
- (8) SSPL and delayed PL spectra and lifetime decay profiles of **PAAQA** were measured in **Supplementary Fig. 35**.
- (9) Lifetime decay profiles and relevant parameter of **PAMCz**, **PAMCzQA** and **PAMCzQAQS** were updated in Supplementary Information (**Supplementary Fig. 38 and Supplementary Table 4**).
- (10) Detailed procedure for the fabrication of hyperafterglow panel was updated in Supplementary Information.
- (11) EL spectra and current density-voltage-luminescence curves of red hyperafterglow LED were updated in **Supplementary Fig. 39**.
- (12) The operation principle of circuitry-controlled LED array was updated in Supplementary Information (**Supplementary Fig. 42**).

All those updates have been highlighted in the revised manuscript and revised Supplementary Information.

If there are any further queries, please feel free to contact me. I can be reached by E-mail: iamrfchen@njupt.edu.cn or by Fax: (+86) 25 8586 6396.

Thank you very much.

Yours sincerely,

Runfeng CHEN

State Key Laboratory for Organic Electronics & Information Displays (KLOEID)

Nanjing University of Posts & Telecommunications

9 Wenyuan Road, Nanjing 210023, China

Tel/Fax: +86 25 8586 6396

Cell: +86 15366190470

E-mail: iamrfchen@njupt.edu.cn

Reviewers' Comments:

Reviewer #1:

Remarks to the Author:

The authors well addressed my concerns. I recommended the publication.

Reviewer #2:

Remarks to the Author:

The authors have addressed my comments and requests and I believe that the paper can now be published.

Reviewer #3:

Remarks to the Author:

The authors addressed all the comments properly and the manuscript can be published in Nature Communications at its current form.

Reviewer comments:

Reviewer: #1

1. The authors well addressed my concerns. I recommended the publication.

Author reply: We appreciate the reviewer's acceptance and recommendation of our work!

Reviewer: #2

1. The authors have addressed my comments and requests and I believe that the paper can now be published.

Author reply: We are very grateful for the recommendation!

Reviewer: #3

1. The authors addressed all the comments properly and the manuscript can be published in Nature Communications at its current form.

Author reply: We appreciate the reviewer's recommendation of our work!